# Atmospheric condition identification in multivariate data through a metric for total variation

Nicholas Hamilton

National Renewable Energy Laboratory, Golden, Colorado, USA

**Correspondence:** Nicholas Hamilton (nicholas.hamilton@nrel.gov)

**Abstract.** Identification of atmospheric conditions within a multivariable atmospheric data set is a necessary step in the validation of emerging and existing high-fidelity models used to simulate wind plant flows and operation. Atmospheric conditions relevant for wind energy research include stationary conditions, given the need for well-converged statistics for model validation, as well as conditions observed less frequently, such as extreme atmospheric events, which are used in wind turbine and wind plant design. Aggregation of observations without regard to covariance between time series discounts the dynamical nature of the atmosphere and is not sufficiently representative of atmospheric conditions. Identification and characterization of continuous time periods with atmospheric conditions that have a high value for analysis or simulation sets the stage for more advanced model validation and the development of real-time control and operational strategies. The current work explores a single metric for variation of a multivariate data sample that quantifies variability within each channel as well as covariance between channels. The *total variation* is used to identify conditions of interest that conform to desired objective functions, such as stationary conditions, ramps or waves of wind speed, and changes in wind direction. Total variation is somewhat sensitive to the presence of outliers in the input data, and the method is best complemented by quality control procedures to ensure reliable results. The direct detection and classification of events or conditions of interest within atmospheric data sets is vital to developing our understanding of wind plant response and to the formulation of forecasting and control models.

## 1 Introduction

Parsing multivariate data sets that are ever growing in size and complexity can be a daunting task for researchers seeking to identify periods or events of interest in time series data (Preston et al., 2009; Shahabi and Yan, 2003). This is especially true for wind energy research seeking to validate high-fidelity numerical models against field observations (Barthelmie et al., 2015; Larsen et al., 2013; Sørensen and Shen, 2002). Wind plants operate continuously over time periods spanning years and across a broad range of atmospheric conditions, each of which implicitly impact the operation of the wind plant, either in terms of power production, operations and maintenance costs, or energy forecasting for grid integration.

Field observations of wind plants are typically collected by instrumentation mounted to wind turbines or meteorological towers, met masts, and by supervisory control and data acquisition (SCADA) systems. Wind plant data sets typically include measurements of wind speed and direction, local temperature and pressure, and wind turbine operational data, such as op-

erational status, power production, and nacelle position. Each of the atmospheric quantities of interest may be classified as non-ergodic stochastic variables that are fundamentally connected (i.e. strongly interdependent).

Wind speed ramps are of particular interest in wind plant power forecasting due to the need to balance energy production against demand curves and in the planning of required reserves and base loads (Sevlian and Rajagopal, 2012; Zhang et al., 2014). Previous work has focused on forecasting of mesoscale changes in wind speed (Bossavy et al., 2013; Ferreira et al., 2011), generally concentrating on risk and reliability issues for wind turbines. Ramp event detection has been a research focus for more than a decade, (Cutler et al., 2007; Ferreira et al., 2013; Hannesdóttir and Kelly, 2019), and has produced some specific recommendations for individual turbine controls and the influence on operations and maintenance costs or activities. Previous research in wind speed ramps is not easily generalized to the identification and characterization of other dynamical events of interest, despite parallels in the detection process and considerations for wind turbine or plant operations and controls.

Detection of events in noisy data is of particular interest in the case of turbulent atmospheric data sets, especially given the need for more sophisticated forecasting systems (Belušić and Mahrt, 2012; Fulcher, 2018; Gamage and Hagelberg, 1993; Kang et al., 2014, 2017; Sun et al., 2015). One of the more common event detection methods leverages the continuous or discrete wavelet transform (Gamage and Hagelberg, 1993; Kumar and Foufoula-Georgiou, 1997; Lilly, 2017). Wavelet transforms leverage time-frequency signals designed to have specific properties that make them easy to use in signal processing applications. However, wavelet transformation remains computationally intensive and requires a fair amount of expertise to implement effectively and avoid the common pitfalls of signal shift sensitivity and the poor representation of phase and directionality (Taswell, 2001). A more direct method simply considers the covariance matrix of the input data, which represents the statistical spread of each data channel as well as cross-correlated variability (Eaton, 1983; Wasserman, 2013). Reducing the variability of a sample of multi-dimensional observations to a single metric is a necessary step to using numerical methods such as least-squares minimization for event detection and classification. Another method for parsing atmospheric conditions found in the literature leverages the Hilbert transform, which convolves time series signals with a Cauchy kernel and results in a phase-shifted set of Fourier components. This method has been used successfully to relate ocean wave conditions to atmospheric conditions through the use of a reference signal (Hristov et al., 1998) and has successfully been extended to turbulence modeling (Kelly et al., 2009; Sullivan et al., 2000) and to relate turbulent motions of various scales within the atmospheric boundary layer (Mathis et al., 2009).

Simultaneous observation of multiple thermodynamic and kinematic quantities reported by met masts are necessary to characterize the dynamical state of the atmosphere (Barthelmie et al., 2014; Hansen et al., 2012). Directly considering multiple disparate data channels simultaneously represents a challenge in that each quantity has different engineering units and that variation within each channel may occur over a distinct scale. Atmospheric conditions are frequently characterized by considering wind speed, wind direction, and turbulence intensity or thermal stability, each of which have different units, ranges, and statistical properties. Consideration of these variables independently may not provide a complete picture of the state of the atmosphere, as they are inherently correlated (Holtslag and Nieuwstadt, 1986; Kaimal et al., 1976); each variable offers a limited range of insights as to the dynamical state of the atmosphere relevant to the operation of wind energy assets. Direct comparison of the marginal distributions of atmospheric variables aggregates observations without regard to the value of other, potentially

correlated variables. Even the use of conditional statistical distributions or measures discounts any dynamic coupling between them and may not fully describe the nature of the atmospheric physics (Hannesdóttir and Kelly, 2019; Preston et al., 2009; Shahabi and Yan, 2003).

The following work explores an application of numerical analysis methods to atmospheric data to identify continuous periods of interest within met mast time series data. The source of the data and their treatment are discussed briefly, although the wind plant and met mast are not in themselves imperative to the demonstration of the method or its utility. A discussion of aggregate statistical measures of the data is followed by a formal definition of the total variability of a block of time series data, and applications using the total variation as a metric to identify specific dynamical events of interest. Alternate metrics exist that quantify the variability of multiple samples or multivariate data. The metrics total variability, overall variability, and summative variance in common use have slightly definitions and interpretations from the total variation introduced in the current work. Briefly, *total variability* is defined as the sum of squares total of difference between expected or mean value and observed qualities. *Overall variability* refers generally to the variance or standard deviation of a population (i.e. a group of samples considered together). *Summative* or *pooled variance* refers to the inferred variance of a population of observations from the collection of sample variances. In contrast, the total variation used in the current work reduces the covariance between normalized variables to a single value through the determinant of the covariance matrix. A close analog to this method is the generalized variance of a multi-dimensional random vector. Generalized variance was introduced by Wilks (1932) and Sengupta (2004) as a scalar measure of overall multidimensional scatter. However, in most formulations of generalized variance, the data are considered as a $p-$dimensional vector. The current work uses the same mathematical operations but applies them to distinct variables that have been merged into a matrix. Mechanically, the same operations are being applied to the data, but given the distinction in formulation, the current work adopts the jargon of 'total variation'. Finally, sensitivity of the method to outliers is analyzed, ending with a discussion of broader applications and extensions to the method.

## 2 Data and quality control

Data used to demonstrate the current method for detecting conditions of interest issue from met mast signals at the Lillgrund Wind Farm, located 10 km off the coast of southern Sweden in the Kattegat Strait. Lillgrund is comprised of 48 Siemens SWT-2.3-93 wind turbines and has a rated nameplate capacity of 110 MW. The layout of the Lillgrund wind plant is shown in Fig. 1(a), where each turbine location is denoted with a marker whose color is representative of the average power produced over the time period analyzed below. Operational data (SCADA, power production, turbine availability) from the wind farm are not discussed further in the following analysis, although a brief summary of future applications of the method is provided in the conclusions section, including thoughts on wind plant performance and SCADA data. Data used to demonstrate the calculation of total variation and identify periods of interest come from the met mast, located at the southwest corner of the wind plant, indicated in Fig. 1(a) with an open marker.

Within any wind plant data, conditions of value for validation are typically identified by way of aggregate statistical metrics or by identifying "well-behaved" time periods exhibiting a dynamical event or atmospheric condition of interest. Kinematic

and thermodynamic atmospheric quantities that are expected to have the greatest impact on the performance of a wind plant are the wind speed $u$, wind direction $\theta$, and the atmospheric stability, considered either in an instantaneous or time-averaged sense. The stability of the atmosphere (typically quantified by the Monin–Obukhov stability parameter or the Richardson number) indicates the magnitude of buoyant production or destruction of turbulent kinetic energy (TKE) relative to shear production of

TKE, and whether it represents either a source or sink of (vertical) momentum (Kumar et al., 2006; Wyngaard, 2010). Forcing in the momentum equations as indicated by the presence and sign of a buoyancy term is manifested in atmospheric flow as vertical turbulent mixing, and is an important overall factor in the energy balance relevant to wind plant operation. Thermal stability has a significant effect on atmospheric turbulence and the structure of wind turbine wakes, wake interaction, and thus the overall energy balance within the wind plant (Ali et al., 2019).

Data used in the current work does not contain any observations of the temperature or heat flux between the atmosphere and the ocean surface, and thus no estimate for the traditional stability metrics are available. Turbulence intensity ($TI$), although an imperfect proxy of atmospheric stability from a fluid mechanical or atmospheric perspective, provides some sense of the energy contained in the fluctuating flow field, and is well-suited for presenting the utility of the total variation method below. Additionally, $TI$ is a quantity frequently used in the wind energy community to characterize wind plant operating conditions

and structural loading of wind turbines (Dimitrov et al., 2018; Kelly et al., 2014) and is often accessible through instrumentation on met masts or wind turbine nacelles making it an appropriate choice for the current demonstration.

Raw data used to demonstrate the current methods include high-frequency (20 Hz) observations of $u$ and $\theta$ reported by the met mast between March and December 2009. Wind speed and direction data were binned to a temporal resolution of 1 min, from which mean and standard deviations were calculated. Turbulence intensity in each bin is estimated as the ratio of the

retained 1-min statistics for wind speed as $TI = \sigma_u/u$. As with most field observations, data availability from each channel is less than 100%, as instruments require maintenance, loose connectivity to data acquisition systems, or shut down to prevent damage under certain conditions. Binning the data into 1-min periods smooths the observed time series of wind speed and direction, and reduces the noise reported by the cup anemometer and wind vane.

Additional quality-control steps for the data include omitting any 1-min period any of the data channels are not correctly

reported from further consideration. Any time stamp associated with wind speeds less than 1 m/s, when wind speed observations reported by cup anemometers and wind vanes are not considered to be reliable (IEC, 2005a), are also removed from the data set. Fig. 1(b) shows data availability of the record as a percent of the total number of data possible per day. The final quality-control step implemented for the current study is to exclude data that are not part of any continuous set of observations of at least 60 min. The current method searches continuous data samples to identify atmospheric conditions and events of

interest. Rather than infill or interpolate data, periods with missing values are simply excluded from consideration.

## 3 Statistical view of atmospheric conditions

Characterization of the atmospheric conditions is most often pursued through aggregate statistics, that is without explicitly considering their evolution in time. Statistical quantities (arithmetic mean values, variances, and higher-order moments) may

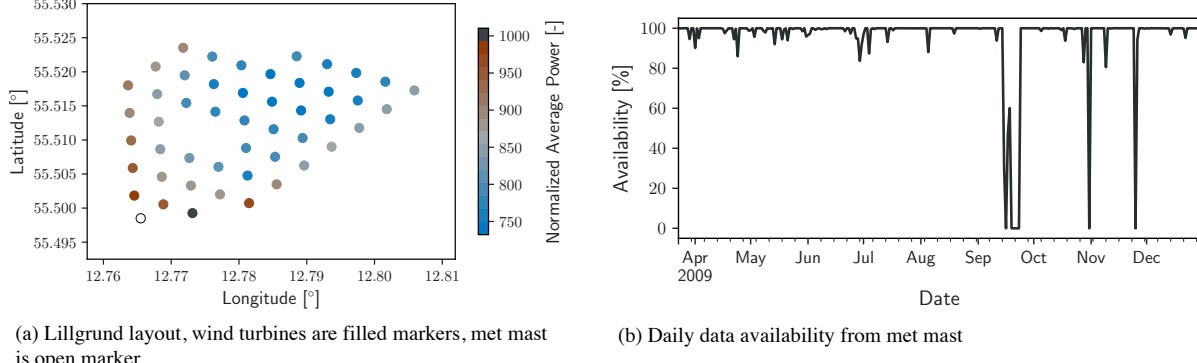

(a) Lillgrund layout, wind turbines are filled markers, met mast is open marker

(b) Daily data availability from met mast

**Figure 1.** Wind turbines, met mast, and data availability from Lillgrund wind plant

reflect the occurrence of infrequent events, but do not convey dynamical evolution of variables or their correlation in time. Considering atmospheric variables in terms of either their marginal distributions (as in Fig. 2) or their conditional distributions (as in Fig. 3) falls short of saying anything about the dynamics embedded in those observations. For example, many steady-state and analytical wake models are defined to represent the time-averaged flow behind a wind turbine and many uses of high-fidelity models assume that the bulk flow speed and direction do not change in time. Effective validation of numerical modeling tools for wind energy requires that observations conform to stationary atmospheric flow (Chenge and Brutsaert, 2005; Metzger et al., 2007; Vincent et al., 2010, 2011; Guala et al., 2011) or represent a dynamic event of interest. Histograms of each of the data channels are provided in Fig. 2, showing characteristic behavior for the wind speed and turbulence intensity distributions.

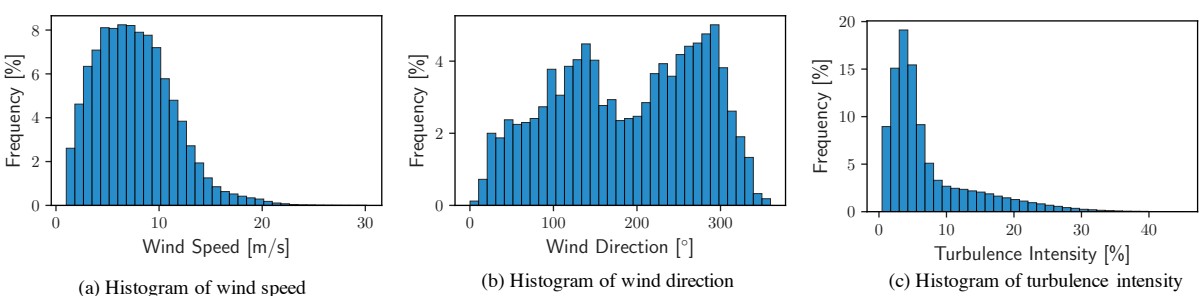

(a) Histogram of wind speed

(b) Histogram of wind direction

(c) Histogram of turbulence intensity

**Figure 2.** Histograms of quality-controlled met mast data

The wind direction (Fig. 2(b)) shows several key features typical of atmospheric records; first, it identifies the prevailing wind directions as per the number of observations within each direction sector (10°) and, second, it shows that virtually no observations correspond with wind directly out of the north. According to the IEC (2005b) Standard for Power Performance Measurements of Electricity Producing Wind Turbines, met masts should be placed sufficiently far from the nearest upstream

obstacle, or risk introducing bias and increased uncertainty into the record. This limitation can be difficult or prohibitively expensive to accommodate due to logistical constraints, especially in offshore settings where placement is often strictly limited.

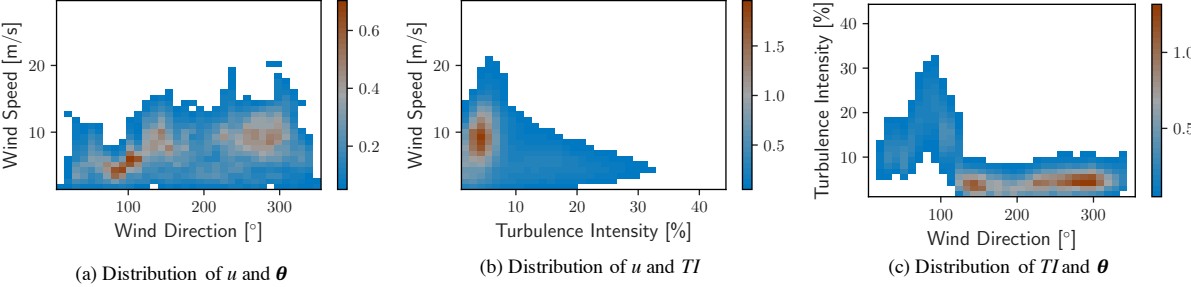

(a) Distribution of $u$ and $\theta$  (b) Distribution of $u$ and $TI$  (c) Distribution of $TI$ and $\theta$

**Figure 3.** Two-dimensional histograms of met mast data. Color information conveys percent of total observations for each pair of variable values.

Each of the histograms in Fig. 2 categorizes a single quantity without regard to the variation of the others; each single-variable histogram effectively integrates the observations over the other two variables. More complex treatment of the data is
required to take into account the simultaneous variability of more than one channel. Fig. 3 shows two-dimensional histograms with two-way permutations of the data channels. In each of the histograms, a threshold has been applied to the frequency of observations. Any bin representing less than 0.5% of the total observations has been filtered out to highlight more common conditions. Two-dimensional histograms demonstrate that the atmospheric conditions are more complex than is possible to estimate from pairwise consideration of any two of the one-dimensional histograms in Fig. 2. An observation from the two-
dimensional histograms that is not immediately evident in one-dimensional histograms is that the greatest turbulence intensity comes from a single, distinct sector of wind directions. Placement of the met mast with respect to the wind turbines contributes to a sharp increase of $TI$ in the range of 15–45% and is not typical of unobstructed measurements. Reports of high $TI$ likely result from the introduction of turbulence to the flow by the wind turbines or wind plant from directions between $70°$–$110°$.

Wind speed and $TI$ roses contain the same information as the two-dimensional histograms from Fig. 4, but convey it on a
polar projection representative of the compass, thus making them more intuitive to read for many users. Fig. 4 shows wind and $TI$ roses for the considered data. The rose diagrams highlight directional dependence of the mapped variable. For example, Fig. 4(b) demonstrates that the greatest turbulence intensity is highly correlated with winds from the sector of $70°$–$110°$. This is the range of directions in which the met mast is waked by the wind turbine located to the west.

## 4   Total variation of dynamical data

Aggregate statistical representation accounts for interdependence of the three variables considered in the current example, but cannot account for the dynamic nature of the atmosphere. A histogram, as a consequence of its composition, only denotes how frequently a given condition is observed without regard to what condition may precede or follow. The actual weather conditions

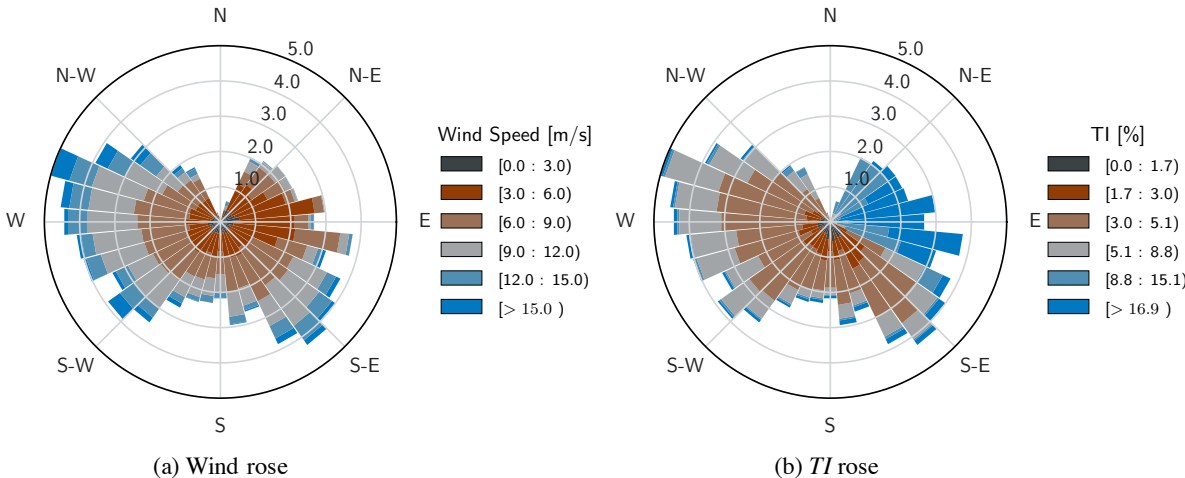

**Figure 4.** Wind (a) and $TI$ (b) roses from met mast data

could well be undergoing a dramatic change, but within any 1-min observation, the variables of interest fall within the stated bounds of a single bin within the full condition space.

An alternate path toward identifying conditions of interest for model validation or benchmarking studies comes through seeking continuous periods from the time series of observations that has properties of interest for a given study. An obvious
choice would be a continuous period in which the atmospheric conditions remain statistically stationary. Statistical stationarity (i.e. time-independence of statistical quantities) is a common consideration in turbulence and atmospheric science (Chenge and Brutsaert, 2005; Metzger et al., 2007; Vincent et al., 2010, 2011; Guala et al., 2011). Stationarity is not often assumed for wind energy research and modeling applications, although it is rarely quantified or even considered in validation data. Additionally, retaining a time series allows users to leverage the interdependence of the channels within a data set by way of correlation or
covariance metrics.

Quantifying the variability of a set of data must include the correlation between data channels, or risk discounting any information regarding the relationship between variables. Stated otherwise, any metric that combines the variability of each channel independently without accounting for covariance between the channels is incomplete and will not be sufficient to fully characterize the state of a given system. Therefore, a method that accounts for variation within each channel and the covariance
between variables is necessary to quantify the distribution of data across multiple channels into a single metric.

Below, each data block, $\mathbf{D}$, is a selected time period and corresponds to an array of size of $[m,n]$, where $m$ is the length of the time period — either 60 or 120 min —and $n$ is three, corresponding to the number of variables $u$, $\theta$, and $TI$.

$$\mathbf{D} = [u(t),\ \theta(t),\ TI(t)] \tag{1}$$

In order for the variability of each channel in $D$, and their respective covariances to be given equal weight, the data must be
normalized to a single common range. Each variable has been normalized by its standard deviation and mapped to an interval

determined by the range of each channel in standard deviations according to the formulation,

$$\mathbf{D}_{\mathrm{norm}} = \frac{\mathbf{D} - \overline{\mathbf{D}}}{\sigma_{\mathbf{D}}} \tag{2}$$

In Eq. (2), the arithmetic mean and standard deviation (denoted by the overline and $\sigma$, respectively) are calculated separately for each column of $\mathbf{D}$. Normalizing data before calculating the total variation ensures that each data stream is weighted equally in the characterization of a given condition or state.

In addition to the definition of $\mathbf{D}$, a block, $\mathbf{f}$, containing objective functions of interest to apply to each of the variables in $\mathbf{D}$ is defined as,

$$\mathbf{f} = [f_u(t),\ f_\theta(t),\ f_{TI}(t)] \tag{3}$$

The difference between objective functions and their respective data will be referred to as a regularized data block, and is noted with a caret,

$$\hat{\mathbf{D}} = \mathbf{D} - \mathbf{f} \tag{4}$$

The purpose of defining an objective function block is to tune the data to show covariance specifically with respect to a desired form about which the data are regularized. Seeking stationary conditions in which minimal variation occurs in all data channels without regularization amounts to the special case of setting the function block to $\mathbf{f} = 0$ (or, more generally, when the objective function is any constant value; $\mathbf{f} = c$). The objective function block is discussed in greater detail in the following sections.

The total variation, $\mathcal{V}$, of a system is a unitless metric to quantify spread of a set of interdependent variables that accounts for autocorrelation within each channel and for covariance between channels. A covariance matrix is calculated for a subset of the data, representing a continuous period of a specified duration,

$$\mathbf{C} = \left(\frac{1}{m-1}\right) \hat{\mathbf{D}}^T \hat{\mathbf{D}} = \left(\frac{1}{m-1}\right) \begin{bmatrix} \sigma_u^2 & \sigma_u \sigma_\theta & \sigma_u \sigma_{TI} \\ \sigma_\theta \sigma_u & \sigma_\theta^2 & \sigma_\theta \sigma_{TI} \\ \sigma_{TI} \sigma_u & \sigma_{TI} \sigma_\theta & \sigma_{TI}^2 \end{bmatrix} \tag{5}$$

In Eq. (5), $\mathbf{C}$ is a square matrix of size $n \times n$ representing the covariance between any pair of data channels. The total variation, $\mathcal{V}$, of a given regularized data block, $\hat{\mathbf{D}}$, is expressed as the determinant of the respective correlation matrix,

$$\mathcal{V} = \det(\mathbf{C}) \tag{6}$$

Larger values of $\mathcal{V}$ indicate that the data points are more dispersed in the condition space. In the observational data of the atmosphere discussed here, $\mathcal{V} > 0$. The case of $\mathcal{V} = 0$ would indicate that the full $n-$dimensional condition space is not occupied and some of the variables are perfectly correlated with, i.e. linearly dependent on, some of the others. Metrics of the variation of a multivariate data set have some history in the literature. Notable past contributions include the pooled variance method to estimate population variance from those of distinct samples (Ruxton, 2006), and the 'total' or 'overall' variability

(Anderson, 1962; Goodman, 1968) which combine variances of individual variables either linearly or in a sum of squares sense. The generalized variance (Wilks, 1932; Sengupta, 2004), shares a common formulation with $\mathcal{V}$, but has historically been applied to a $p-$dimensional random vector. In contrast, the total variation merges $n$ distinct variables, whose relationship need not be known a priori, and seeks the determinant of the associated correlation matrix.

## 4.1  Quiescent conditions: $\mathbf{f} = \mathbf{c}$

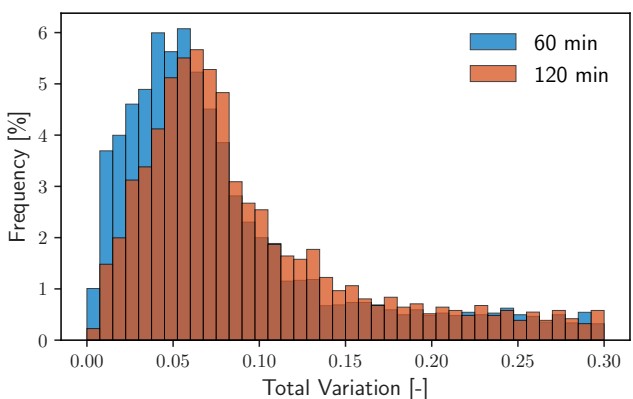

**Figure 5.** Distribution of $\mathcal{V}$ for data blocks of 60 or 120 min (blue and red, respectively)

Fig. 5 shows the distribution of $\mathcal{V}$ dividing the data record into continuous periods of either 60 (blue) or 120 min (red). Both distributions in Fig. 5 have been limited to $\mathcal{V} \leq 0.30$ to emphasize differences between the two data block lengths. In either case, the distribution is positively skewed, and high values of $\mathcal{V}$ exist with very low frequency. Immediately visible in the histograms of $\mathcal{V}$ is that there is a range of values exhibited most commonly by the blocks of data. For data broken into 60-min periods, 35.9% of blocks have a total variation less than 0.05, whereas for data broken into 120-min periods, only 25.0% of blocks have a total variation in the same range. Although $\mathcal{V}$ is a unitless metric, its relative value does convey the degree of variation represented by all data within a respective time period. The values of $\mathcal{V}$ with the greatest frequency of occurrence is larger for periods of 120 min than for periods of 60 min. This is an expected trend because of the greater changes in atmospheric conditions that are possible within a larger window. There remains an inherent trade-off between the length of a data block and the degree of variation; longer blocks provide greater statistical convergence of $\mathbf{C}$, but risk including more dynamical variation, which contributes to higher values of $\mathcal{V}$.

Periods of time corresponding to the minimum values of $\mathcal{V}$ are those in which the total atmospheric conditions vary the least. In these periods, small values of standard deviation within each data channel as well as minimal covariance between the channels is expected. Minimal covariance between channels is equivalent to observing only stochastic, uncorrelated fluctuations in each channel. In contrast, periods corresponding to the maximum values of $\mathcal{V}$ are those in which the subset of data experiences the greatest variability, to which individual channel noise and correlated events between channels both contribute. Time periods of 120 min corresponding to the maximum (red) and minimum (blue) total variation are shown in Fig. 6(a). To

provide a broader sense of how other time periods are characterized in terms of $\mathcal{V}$, five randomly selected periods of 120 min are shown in Fig. 6(b). The principal components of each data block are shown with black vectors and the total variation is listed in the legend. The figure represents each block of data as a scatter of only normalized wind speed and direction, although $TI$ is also in the calculation of $\mathcal{V}$.

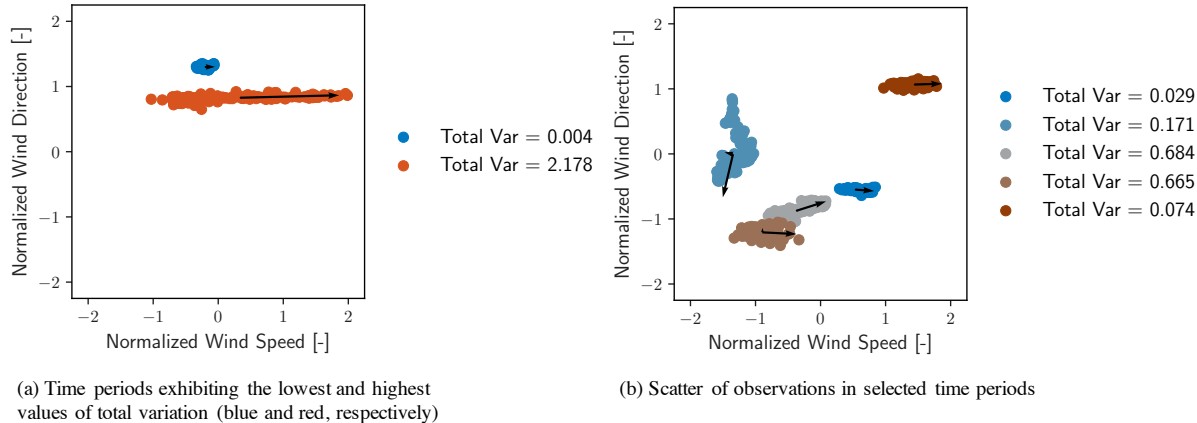

(a) Time periods exhibiting the lowest and highest values of total variation (blue and red, respectively)

(b) Scatter of observations in selected time periods

**Figure 6.** Scatter of data points of selected time periods within the full conditions space

Fig. 7 shows the wind speed, direction, and turbulence intensity corresponding to the 10 periods of minimum and maximum total variation. Each variable is shown in its original (non-normalized) engineering units to provide insight into the atmospheric conditions, although they were identified using normalized data. Fig. 7(a) shows that the periods with minimal values of $\mathcal{V}$ have time series that appear constant and experience only small stochastic variations within each channel and that periods with large values of $\mathcal{V}$ exhibit more spread. For each set of time series, the extreme values are shown in the boldest color (red, blue, and

gray for the wind speed, direction, and turbulence intensity, respectively) and fade to lighter colors for more moderate values of $\mathcal{V}$. Starting and ending times are not included, as Fig. 7 is intended only to demonstrate the sorting capability of the method.

## 4.2   Objective conditions: f $\neq$ 0

Regularizing the data with respect to a set of nonzero objective functions centers the of $\mathcal{V}$ around specific conditions of interest. For example, in the case of wind plant analysis, it may be of interest to assess array performance during a wind speed ramp

event or change of wind direction. Such events may be readily formulated according to accepted mathematical definitions and supplied to the total variation algorithm from Section 4. Defining specific objective functions will quantify the total variation around those conditions, which can then be used to identify the time periods that match the event of interest most closely.

    An additional step is considered to sort the full data set for a more general formulation. In such a case, events of interest are defined in a suitably general formulation, and a least-squares minimization is applied to seek the relevant parameter values.

In the current demonstration, function types of interest are wind speed ramps, wind speed waves, and wind direction changes,

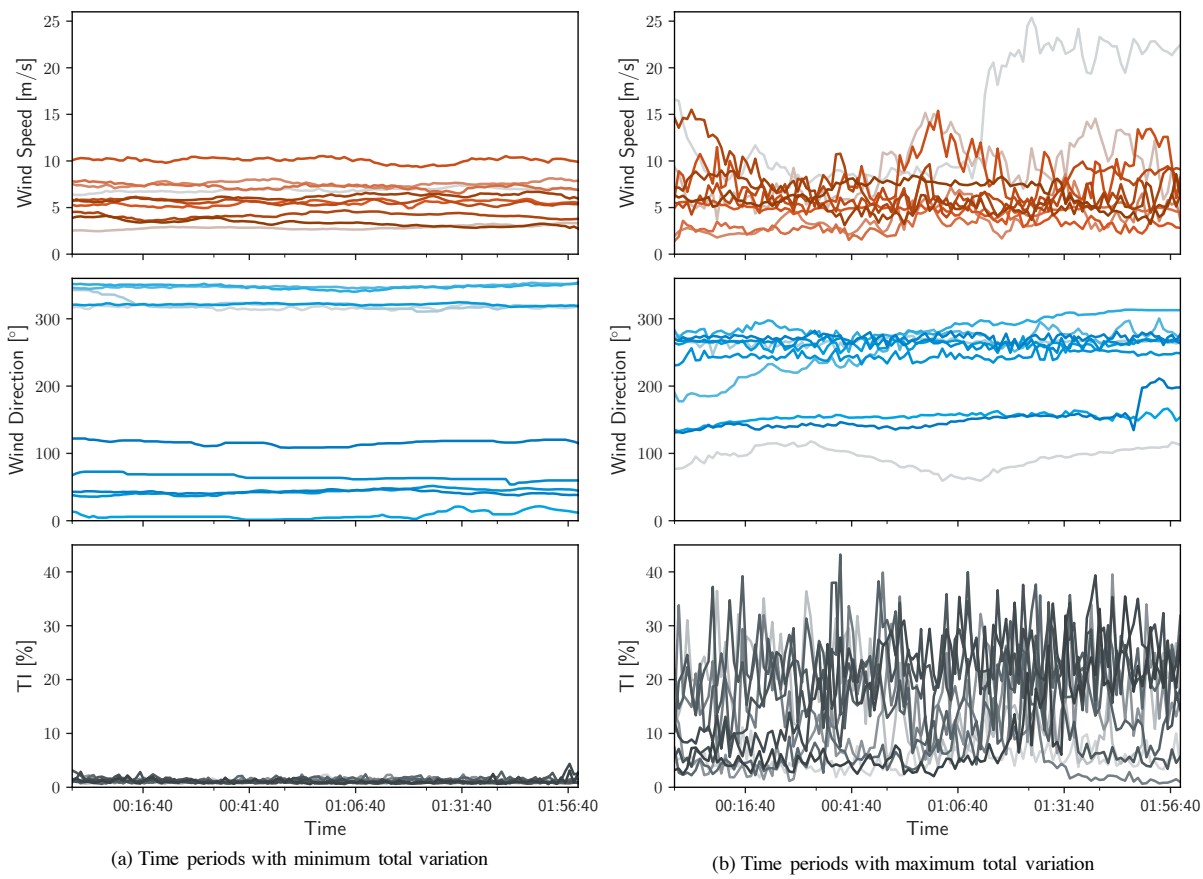

(a) Time periods with minimum total variation

(b) Time periods with maximum total variation

**Figure 7.** Time series of the 10 blocks with minimum and maximum values of $\mathcal{V}$, (a) and (b), respectively

shown in the function blocks Eqs. (7), (8), and (9), respectively, distinguished with the subscripts $A$, $B$, and $C$.

$$f_A = \begin{cases} f_u(t) &= c_0\,t + c_1 \\ f_\theta(t) &= 0 \\ f_{TI}(t) &= 0 \end{cases} \tag{7}$$

$$f_B = \begin{cases} f_u(t) &= c_0 \sin(c_1\,t + c_2) + c_3 \\ f_\theta(t) &= 0 \\ f_{TI}(t) &= 0 \end{cases} \tag{8}$$

$$f_C = \begin{cases} f_u(t) & = 0 \\ f_\theta(t) & = c_0 \arctan\left(c_1\, t + c_2\right) + c_3 \\ f_{TI}(t) & = 0 \end{cases} \tag{9}$$

In each of the equations for $f_A$, $f_B$, or $f_C$, objective function parameters, $c_i$, are sought through least-squares minimization of the following expressions,

$$5 \quad \rho = \left\| \hat{\mathbf{D}} - \mathbf{f} \right\|^2 = \begin{cases} \min \sum \left(u(t) - f_u(t, c_i)\right)^2 \\ \min \sum \left(\theta(t) - f_\theta(t, c_i)\right)^2 \\ \min \sum \left(TI(t) - f_{TI}(t, c_i)\right)^2 \end{cases} \tag{10}$$

where $\rho$ is the least-squares fit residual. Least-squares fit parameters and the respective fit residual from each time period are retained, enabling an additional layer of filtering for conditions of interest. After objective function coefficients are determined, the total variation method is continued, yielding a value of $\mathcal{V}$ for regularized data in each time period. Regularizing the data block by subtracting away objective functions amounts to "detrending" the data such that the covariance matrix reflects correlation among the remaining data.

Fig. 8(a) compares distributions of $\mathcal{V}$ given the objective function definitions in Eq. (7), (8), and (9). The distributions indicate that the total variation can be reduced by regularizing data around generalized sinusoidal (red), linear (blue), and inverse tangent (black) functions as compared to the case where $\mathbf{f} = 0$ (gray). However, the reduction in $\mathcal{V}$ for the full data set is caused by the general definitions of the objective functions. Defining the coefficient values ahead of time would likely *increase* the average value and spread of $\mathcal{V}$; for example, it is not expected that a wind speed ramp with specific slope and vertical offset would fit every time period well, and thus would not necessarily reduce the total variation for that period.

Noted earlier, the additional step of least-squares minimization provides a fit residual for each time period under consideration, shown in Fig. 8(b). Fit residuals indicate the goodness of fit of a given time period to the specified objective function forms. The distributions in Fig. 8(b) suggest that inverse tangent and sinusoidal functions fit the data with less residual error, $\rho$, than a linear objective function. This is likely caused by the additional objective function parameters (degrees of freedom) available for tuning the minimization.

Adding an auxiliary step to the search process of least-squares minimization to a given objective function quantifies the goodness of fit of each data block and can return the parameter values necessary for the desired fit. For example, a least-squares fit to a linear relationship for any data channel will provide values of slope and offset as well as a residual value indicating the quality of the fit. In this way, the data provide alternative values for which sorting may be applied in addition to the total variation.

Figs 9(a) and 9(b) show a selection of periods with minimal total variation around linear and sinusoidal objective functions of wind speed, corresponding to wind speed ramps and waves, respectively. Selection of the wind speed ramps in Fig. 9(a) are conditioned to have the minimal total variation, minimal fit residual, and maximum absolute values of slope. These are

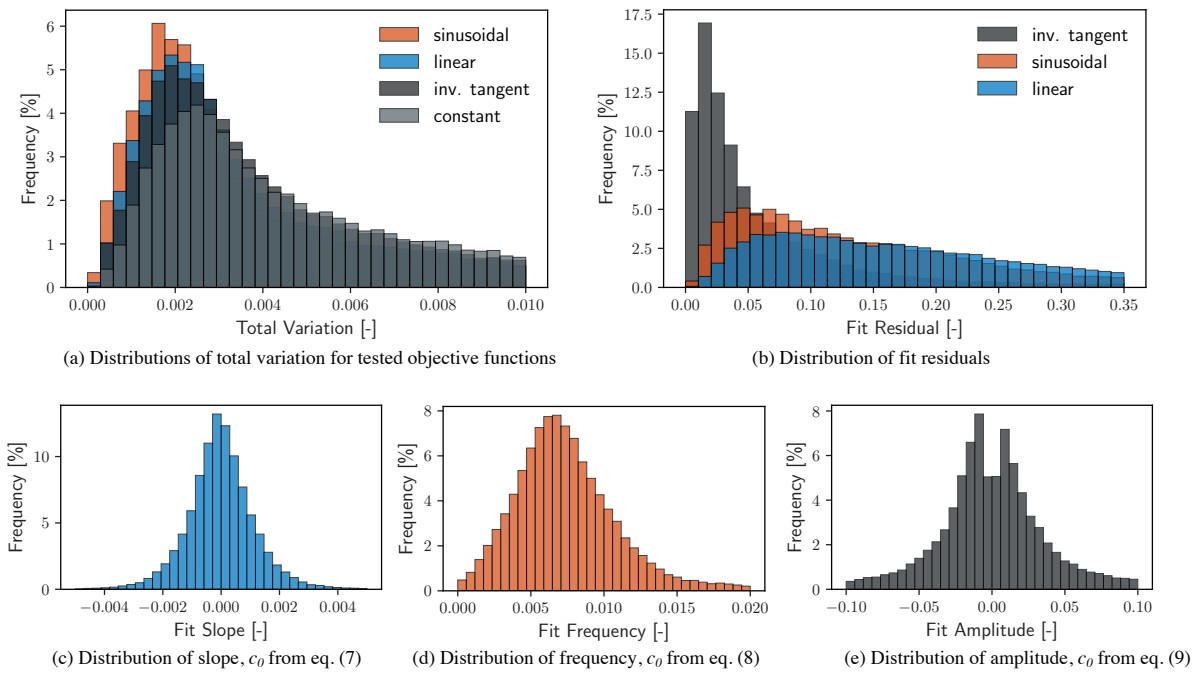

**Figure 8.** Distributions of selected quantities for selected objective functions

the time periods in which the wind speed ramps are simultaneously the most well-behaved (i.e. minimal fit residual) and most intense (i.e. greatest absolute value of slope). Similarly, the wind speed waves shown in Fig. 9(b) were selected by seeking the minimal total variation and then selecting time periods in which the fit frequency fell between desired limits. In Fig. 9(b), the top subfigure shows 120-minute time periods in which the fit frequency is in the range of $[0.015, 0.02]$ rad/s (in red), and the

5   bottom subfigure shows time periods in which the fit frequency is in the range of $[0.0075, 0.008]$ rad/s (in blue). Frequency limits were selected arbitrarily, and are meant only as a demonstration of the method's independence of fit frequency. Fig. 9(c) applies an inverse tangent objective function to the wind direction channel while seeking constant conditions in wind speed and turbulence intensity, identifying the periods of wind direction change with minimal total variation. Direction changes were considered in an absolute sense, and Fig. 9(c) shows time periods with minimal $\mathcal{V}$ in which the absolute direction change $|\Delta\theta|$

10   falls in the range $([20°, 40°]$. Again, the particular magnitude of direction change selected here is arbitrary, and was selected only to demonstrate the fit to an inverse tangent objective function.

## 5   Sensitivity to outliers

A word of caution on using the total variation to identify periods of interest: Because principal component analysis is sensitive to outliers contained in the data, the method may falsely classify a time period as having a large value of total variation due to a

15   few spurious data points. Consideration of outliers in multivariate space requires a similar treatment as for the consideration of

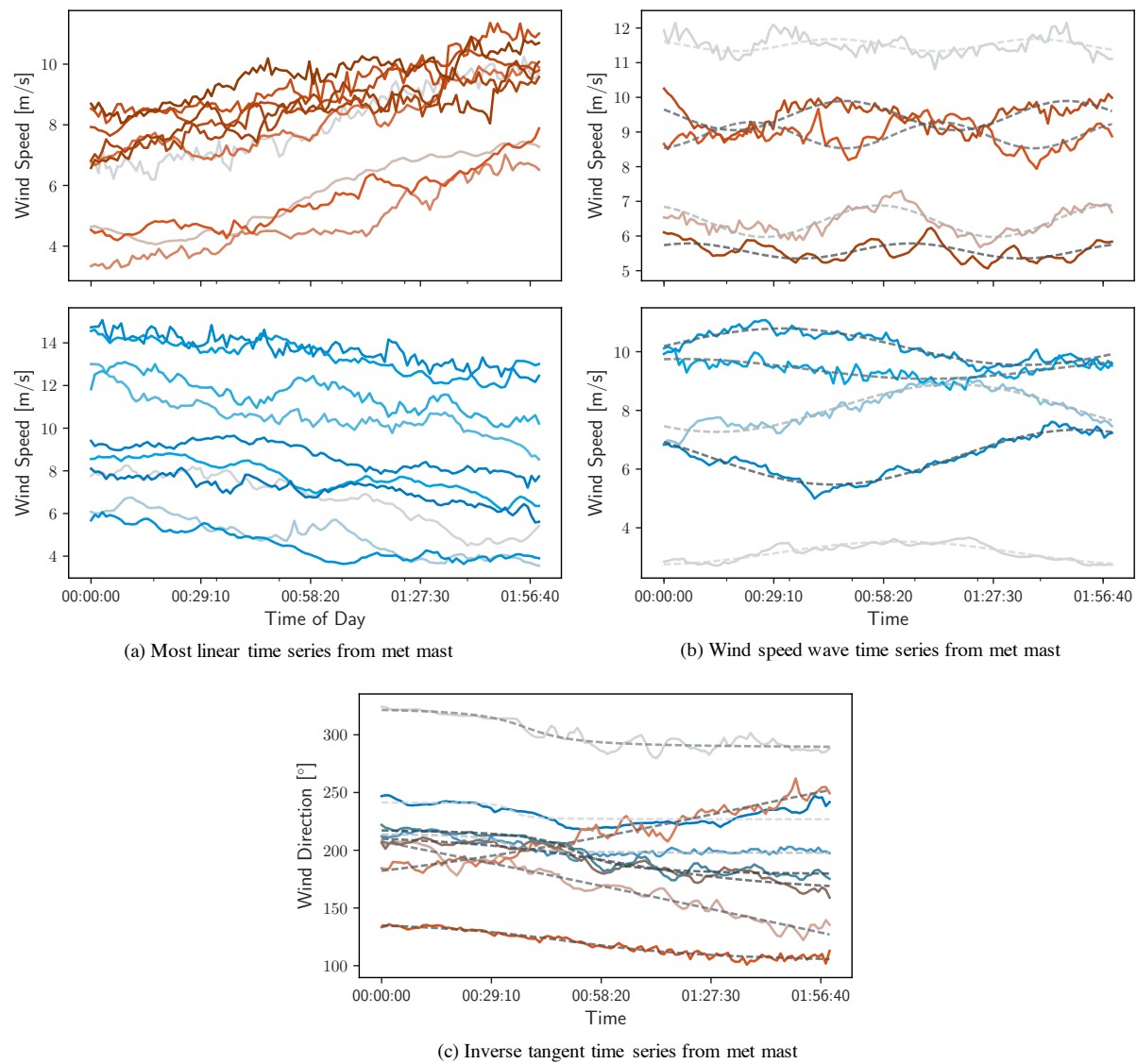

(a) Most linear time series from met mast

(b) Wind speed wave time series from met mast

(c) Inverse tangent time series from met mast

**Figure 9.** Examples of time series identified by calculating covariance matrix around linear, sinusoidal, and inverse tangent objective functions

total variation. Seeking outlying points in each data channel individually discounts the possibility that the other data channels may be within acceptable statistical limits for the same point. Determining outliers from individual data channels further discounts any correlation that may exist between the channels. An effective means of considering outliers in multivariate data is the Mahalanobis distance, $\chi$, which quantifies the Euclidean distance of a point from the center of a data set in terms of

standard deviations (De Maesschalck et al., 2000; Hadi, 1992; Rousseeuw and Van Zomeren, 1990; Xiang et al., 2008),

$$\chi = \sqrt{(x - \mu)^T C^{-1} (x - \mu)} \tag{11}$$

The Mahalanobis distance is sought through the covariance matrix of the data, and thus accounts for interdependence of the data channels, as emphasized earlier. Setting a threshold value for the Mahalanobis distance effectively draws an $n$-dimensional

5  ellipsoidal boundary around the data set in nondimensional space, outside of which data are considered invalid.

   To quantify the sensitivity of $\mathcal{V}$ to the presence of outliers, 10,000 synthetic data sets are generated, and outliers are detected and eliminated. Total variation is compared for each data set before and after outlier detection/elimination. Synthetic data sets ($n$=2 dimensions, 1,000 points each) are normally distributed about a zero mean value with a standard deviation that is randomly assigned in the range of [0, 10]. Each data set is normalized, given a random shape parameter to stretch the data,

10  and rotated to simulate covariance between data channels. The covariance matrix is calculated using Eq. (5) and $\mathcal{V}$ calculated as in Eq. (6). Any point with $\chi > 3$ is flagged as an outlier and eliminated. With two degrees of freedom (variables in the data block), values of $\chi > 3$ are expected to be observed with a probability of approximately 1.1% (Penny, 1996; Ben-Gal, 2005; Gellert et al., 2012). The total variation is then calculated for the cleaned data without outliers, for comparison.

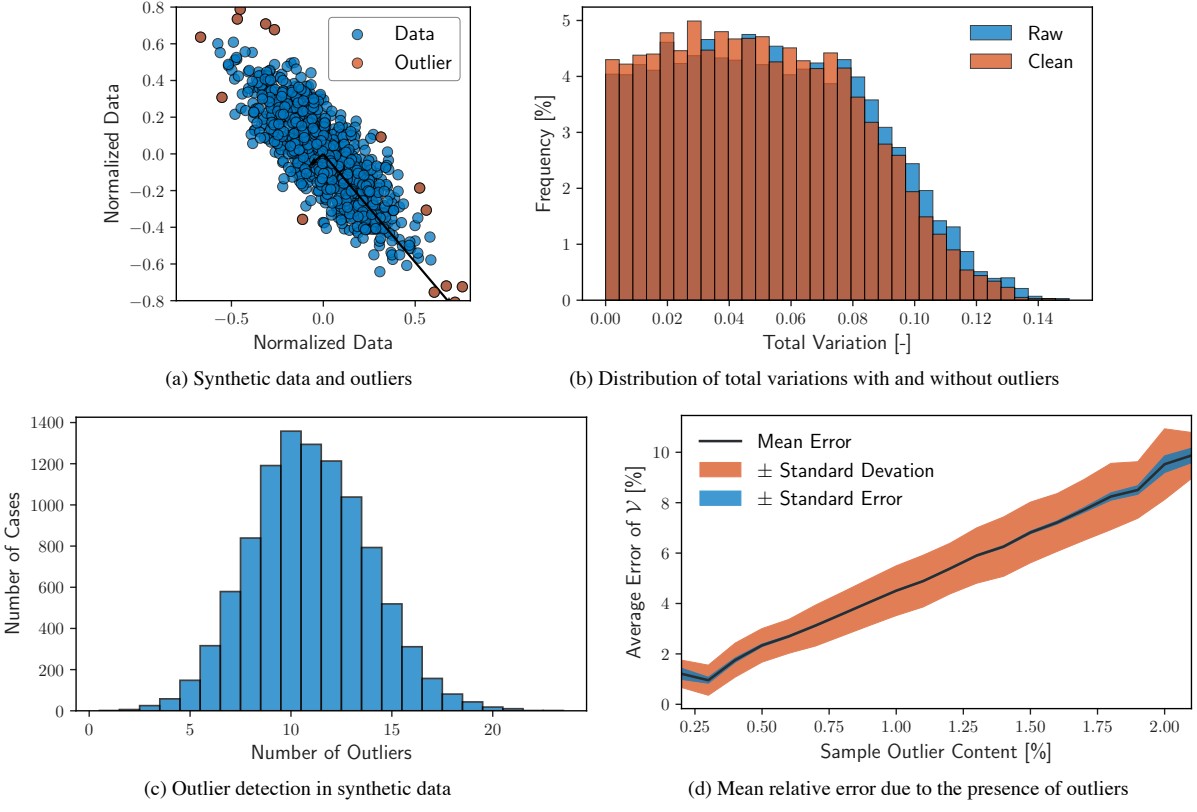

(a) Synthetic data and outliers

(b) Distribution of total variations with and without outliers

(c) Outlier detection in synthetic data

(d) Mean relative error due to the presence of outliers

**Figure 10.** Outlier detection and the sensitivity of $\mathcal{V}$ to outliers

Fig. 10(a) shows a single example set of synthetic data. Accepted data are shown in blue, outliers in red, and the principal components of the data are shown as the black vectors. Fig. 10(b) shows distributions of $\mathcal{V}$ before and after exclusion of outlying data identified with a threshold of $\chi$ in blue and red, respectively. As expected, the total variation of data sets without outliers is smaller than data sets before cleaning. Because of the large number of synthetic data sets considered, statistics regarding sensitivity to outliers are also within reach.

Fig. 10(c) shows the distribution of the number of detected outliers within each synthetic data set. Fig. 10(d) shows the mean relative error according to the number of detected outliers according to

$$\varepsilon = \frac{\mathcal{V}_{\text{raw}} - \mathcal{V}_{\text{clean}}}{\mathcal{V}_{\text{raw}}} \tag{12}$$

where the subscripts denote the presence and absence of outliers (raw and clean, respectively). Uncertainty of the error is shown as the shaded bands around the mean relative error. The red band indicates the standard deviation of the relative error ($\sigma_\varepsilon$) and the blue band denotes the standard error ($\sigma_\varepsilon / N_{\text{outliers}}$). The roughly linear relationship shown in Fig. 10(d) indicates that one could expect an increase in error of approximately 4% for each additional percent outlier content of a given data set.

It should be noted that the present error analysis is not expected to yield identical results for atmospheric data. Observations of wind speed, direction, and turbulence intensity can vary considerably during any given period as part of the normal development of weather patterns. Mentioned briefly in the introduction, quality control of met mast and SCADA data is an active research topic and is beyond the scope of the current method development. However, it should be clear from the sensitivity analysis undertaken here that a careful quality control process should be applied before calculation of the total variation.

## 6    Conclusions

The definition of high-value conditions for wind plant analysis is ultimately up to the user, but may not conform to the most frequently observed state. For example, it may be of greater concern to wind plant developers, owners, or operators to be able to validate models where wake losses are greatest or during ramps of wind speed. These conditions may be more relevant to control or curtailment actions of wind plants, and may have a greater impact on the return on investment of wind energy assets.

Identification of continuous time periods that conform to conditions of interest is not intuitive through aggregate statistics, such as measures of central tendency or even joint probability distributions. The method to quantify the total variation of a multivariate data set described earlier provides a computationally economical means of parsing large and complex data sets, and includes a mathematically robust approach to sorting with respect to a desired condition or objective function. In addition, the method should be equally applicable to any data, regardless of which variables are part of the data block and for data of any length and resolution, provided that enough observations are present to ensure reasonably converged statistics. Normalizing the data makes combining disparate types of data into a single metric possible and meaningful.

The total variation method for seeking conditions of interest has applications far beyond the demonstration undertaken in the current work. Once properly classified, any number of detection and forecasting models may be trained and thoroughly validated. Collecting time periods containing similar dynamical events opens a path forward for more advanced analyses,

such as modal decomposition methods and reduced order modeling. Extreme atmospheric events, as from the International Electrotechnical Commission (IEC) Standard for Wind Turbine Design (IEC, 2005a), have well-defined characteristic functions that could readily be incorporated into the method explored in this article. Extreme atmospheric event definitions can be included in the definition of the function block and used to regularize the data, providing an algorithmic means of identifying

extreme events in historical data records for subsequent analysis.

The total variation method explored here details identification and characterization of time series data from met masts only. Validation of high-fidelity wind plant models frequently relies on some form of operational data, most often power production or some integrated statistic of wind plant performance. SCADA signals and power production or fault events could readily be identified with the total variation method. A further extension of the method would be to add functionality that accounts

for spatial variation of operational data within a wind plant. A spatial aspect to the total variation method would augment the process to be able to detect and characterize the movement of weather fronts through a wind plant or cases in which wake losses are particularly significant and heterogeneous.

An example Python library has been uploaded to a public repository at https://github.com/nhamilto/total-variation (Hamilton, 2020). In the repository also can be found a synthetic data generator used for the analysis of outlier sensitivity and

met mast data similar to that used in the work above. Data in the example is derived from a meteorological mast located at the National Renewable Energy Laboratory's Flatirons campus. Additional data from the met mast is available at https://nwtc.nrel.gov/MetData (NREL, 2020) and the atmospheric conditions at NREL are summarized in Hamilton and Debnath (2019).

*Acknowledgements.* This work was authored by the National Renewable Energy Laboratory, operated by Alliance for Sustainable Energy,

LLC, for the U.S. Department of Energy (DOE) under Contract No. DE-AC36-08GO28308. Funding provided by the U.S. Department of Energy Office of Energy Efficiency and Renewable Energy Wind Energy Technologies Office. The views expressed in the article do not necessarily represent the views of the DOE or the U.S. Government. The U.S. Government retains and the publisher, by accepting the article for publication, acknowledges that the U.S. Government retains a nonexclusive, paid-up, irrevocable, worldwide license to publish or reproduce the published form of this work, or allow others to do so, for U.S. Government purposes. Data was furnished to the authors

under an agreement between the National Renewable Energy Laboratory, Siemens Gamesa Renewable Energy A/S, and Vattenfall. Data and results used herein do not reflect findings by Siemens Gamesa Renewable Energy A/S and Vattenfall. Additional thanks to the National Renewable Energy Laboratory Wake Squad for the musings and dialogue that ultimately led to this work getting started (and finished). Special thanks to Tony Martinez for countless discussions on everything from numerical methods to physical interpretations, editing, and computational assistance with volume rendering. Bridging the gap between my own turbulence experience and Mike Optis' atmospheric

perspective essentially framed the discussion and motivation of the work.

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
