# Peer review of "Atmospheric condition identification in multivariate data through a metric for total variation"

_Atmospheric Measurement Techniques, 2019_

## Referee Comment (RC1) · 12 Nov 2019

General comments: This paper offers a new interesting method to analyse a multivariable atmospheric data set. The method is clearly described, and the data analysis is thorough. It seems like a very versatile method with potentially many different usage scenarios. The manuscript would benefit by considering the points below.

The language could be simplified at times and some sentences could be broken up for easier reading.

Abstract: The problem statement in the abstract could be shortened, while at the same

time there could be more information on the subject/method itself. The abstract would also be improved by including main results and findings.

Sections 2 and 3 could be shortened a bit. This would give the main part (section 4) of the paper more focus. A suggestion: Perhaps Table 1 and corresponding text could be removed as it is not so relevant for the focus in the paper.

It is not completely clear what parts of the method are novel and what has been done before. This could be pointed out.

The paper would benefit from a stronger discussion, perhaps in a dedicated section of its own.

A thought: When we apply specific objective functions, we generally decrease the total variation and find conditions of interest by minimizing the total variance. Would we find similar conditions by not applying any objective functions and maximizing the total variation instead?

Are there any available codes or scripts with this method implemented?

Specific comments:

In the abstract lines 3-4: "Most often, conditions of interest are determined as those that occur most frequently. . ." And similarly, p. 3, lines 12-13: "Within any wind plant data. . ." This statement would benefit from a reference, because it could be argued that the opposite often holds true. E.g. for wind turbine site assessment and certification, the conditions of interest are critical weather and extreme conditions.

Introduction, p. 2, lines 25-27: This is quite a strong statement – it would benefit from a citation or further argumentation. Introduction, p. 2, lines 27-29: Direct comparison of statistical quantities to what? Why does that discount the coupling between quantities that underpin atmospheric physics? This could be clarified and explained further.

Figure 1 a): The numbers on the colorbar are missing the number 9 in front of 00 and

50.

Equation 4-5: Should just be a single equation with one number. Further, I cannot see how the matrix multiplication would result in the covariance matrix. Unless the average of each column has been subtracted from the values in D ÌĆ and the values have been divided by m. If that is the case, it should be mentioned. At this point in the paper normalization has not been mentioned.

Figure 6: It does not seem that the histogram adds up to 100%. Has the data been cut off at Total Variation=0.3? It would be better to show the whole range of the Total Variation.

Figure 7 a) is not mentioned anywhere in the text. It should be commented and explained in the text.

Page 12, equation 12-13. Again, should just be one equation. Also, what objective function is used for the TI? It is not mentioned.

Page 12: When the objective function eq. 9 is applied to the wind speed, what objective functions are then applied to the direction change and TI at the same time?

Page 12, lines 17-18: "Defining specific functions, even of the same forms, would likely increase the average value and spread of V...". Are you certain of this? According to Figure 9 a) the average value and spread of V has decreased by subtracting the objective functions from the data. As you mentioned, subtracting the objective function acts as detrending, and therefore it should be expected that the total variation would always decrease, as it is only the stochastic part of the data that determines the covariance of the remaining data.

Figure 9: Includes two subfigures named (d). Also, these are not mentioned in the text, but should be. What is fit frequency - is it connected to eq. 11? Could you elaborate?

Section 5: The data used in this section is synthetic, and provides a very illustrative example of the sensitivity. However, I wonder if the removed points can be interpreted

as outliers. Could we not say that these are extremes? Maybe the outliers could be assigned standard deviations outside of the range [0, 10], to ensure that they represent "real" outliers due to e.g. measurement errors.

Page 16, line 25: "... the method is independent of the length of the data record...". How can this statement be supported by the current analysis?

---

## Referee Comment (RC2) · Mark Kelly (Referee) · 25 Nov 2019

Please see attached annotated PDF for more review comments.

There is some good content and work here, with a generalized method to find condi-tions of interest for multivariate timeseries; and (perhaps more importantly) inclusion of responsible application of a metric (Mahalonbis distance) to evaluate sensitivity of the method to outliers.

The title is perhaps not quite appropriate; "Total variation of atmospheric data" is rather

vague and somewhat grandiose, not accurately capturing the essence of the work and connoting more results/applicability than demonstrated.

Some significant items of note, as a list:

- In the abstract, *'periods'* of interest is better expressed as '*conditions*', both for the sake of validation and for getting conditional statistics (and towards making fair comparisons of statistics given some conditions).

- Stationarity and conditional statistics underpin this written work; these concepts should be integrated (and referenced, as found in various texts for atmospheric flows), at least starting with the literature review.

- In your literature review, a key method/scheme for event detection (beyond wavelets) appears to be missing: i.e., reference-signal (or ideal signal) approaches based on Hilbert transform, as in Hristov *et al* (1998, PRL **81** no.23), used in various literature (e.g. Kelly, Wyngaard & Sullivan 2009).

- When you mention "direct comparison of statistical quantities", it appears that you are trying to refer to statistics based on *marginal distributions* (or marginal statistics), are you not? In statistical parlance, one contrasts between marginal and conditional statistics.

- The premise "*In lieu of a time series of Richardson number or the Monin-Obukhov stability parameter, turbulence intensity (TI) is used in the current demonstration as a proxy for stability*" is fundamentally problematic. That is, the balance of mechanical (shear) production, buoyant production or destruction, and dissipation $\varepsilon$ (defining the 'simple' conditions where Monin-Obukhov similarity applies) results in TI being a proxy for stability only for flows/conditions with the same dissipation rate (Kelly, Larsen, Dimitrov & Natarajan, 2014). So your results per $TI$ are conditional on $\varepsilon$, and do not act as such a proxy unless further constrained (e.g. via $U$ assuming surface-layer similarity for $\varepsilon$.)

Since stability is not really used in the paper, I suggest that you simply keep $TI$, and change the justification for its use: $\sigma_u$ and $TI$ are important for driving turbine loads (e.g. Dimitrov, Kelly, Vignaroli & Berg 2018).

- In section 3, where you write "*without explicitly considering the evolution of atmospheric variables*" you should mention stationarity as well. In the atmospheric sciences and boundary-layer meteorology this is typically considered, whereas it is often neglected in wind energy applications.

- Figure 5: missing axis values/scales

- Section 4: can you interpret the total variation in terms of the multivariate components, to avoid obfuscation? Section 4.0 (p.8) is essentially taken from PCA; you should include reference to appropriate PCA text(s) and try to explain $\mathcal{V}$ for the reader. E.g., for readers not as 'fluent' in statistics, if the PC's ($\mathcal{P}$) are orthogonal, then how are the covariances accounted for?

  Is your $\mathcal{V}$ different than the 'overall' or 'total' variability found in literature?

  It could help also to point out the difference between summative variance and $\mathcal{V}$.

- Figure 8: suggestion: use logarithmic scale on y-axis to compare more sensibly

- Fig.9c: which "dimensionless slope" are you using here?

- Fig.11: captions are swapped between (c) and (d).

Please also note the supplement to this comment:
https://www.atmos-meas-tech-discuss.net/amt-2019-200/amt-2019-200-RC2-supplement.pdf

**Supplement:**

**Total variation of atmospheric data:** covariance minimization about objective functions to detect conditions of interest**

Nicholas Hamilton

National Renewable Energy Laboratory, Golden, Colorado, USA

Correspondence: Nicholas Hamilton (nicholas.hamilton@nrel.gov)

[revised manuscript text omitted]

---

## Author Comment (AC1) · 24 Dec 2019

**Response to Comments from Reviewer 1**

**amt-2019-200 Total variation of atmospheric data: covariance minimization about objective functions to detect conditions of interest Nicholas Hamilton**

General comments: This paper offers a new interesting method to analyze a multivariable atmospheric data set. The method is clearly described, and the data analysis is thorough. It seems like a very versatile method with potentially many different usage scenarios. The manuscript would benefit by considering the points below. The language could be simplified at times and some sentences could be broken up for easier reading.

Thank you for taking the time to add your thoughts to the submitted manuscript. In addressing them, I think you will see that the manuscript has been greatly improved. Below, I have provided a brief response to each of the points you raised in the review of my work and, where appropriate, also included any additions or subtractions from the manuscript. In addition, I have edited the text in the manuscript to simplify the language where possible to increase readability. The manuscript has been greatly improved, due to your comments and the review process.

Abstract: The problem statement in the abstract could be shortened, while at the same time there could be more information on the subject/method itself. The abstract would also be improved by including main results and findings.

Following the suggestion of the reviewer, the abstract has been revised to be more concise, while more clearly communicating the central contribution of the work. Given that the manuscript is focused on the introduction of a method, the abstract now points out the merits of the methods and points toward the sensitivity due to outliers as quantified through the Mahalanobis distance.

Sections 2 and 3 could be shortened a bit. This would give the main part (section 4) of the paper more focus. A suggestion: Perhaps Table 1 and corresponding text could be removed as it is not so relevant for the focus in the paper.

**Sections 2 and 3** detail common steps used in the quality control of the atmospheric data and the aggregate statistical methods for wind energy. I feel that these sections are necessary to properly establish the narrative of the manuscript and to differentiate the total variation method introduced in the paper. As suggested by the reviewer, these sections have been revised where possible to make them more compact, while keeping their content clear and concise.

It is not completely clear what parts of the method are novel and what has been done before. This could be pointed out.

The method relies on the classical understanding of correlated signals common to the analysis of physical systems. There are parallels with previous work as noted in the literature portion of the introduction. To reinforce this in the work, new references have been added for generalized variance, and a statement has been added to distinguish the novel contribution of the method developed in the paper and its application.

"The total variation, V, of a given regularized data block, D, is expressed as the determinant of the respective correlation matrix,

*V=det(C)* (6)

Larger values of Vindicate that the data points are more dispersed in the condition space. In the observational data of the atmosphere discussed here, V>0. The case of V= 0 would indicate that the full n-dimensional condition space is not occupied and some of the variables are perfectly correlated with, i.e. linearly dependent on, some of the others. Metrics of the variation of a multivariable dataset have some history in the literature. Notable past contributions include the pooled10variance method to estimate population variance from those of distinct samples Ruxton (2006), and the 'total' or 'overall' variability Goodman (1968); Anderson (1962) which combine variances of individual variables either linearly or in a sum of squares sense. The generalized variance (Wilks, 1932; Sengupta, 2004), shares a common formulation withV, but has historically been applied to a p-dimensional random vector. In contrast, the total variation merges n distinct variables, whose relationship need not be known a priori, and seeks the determinant of the associated correlation matrix"

The paper would benefit from a stronger discussion, perhaps in a dedicated section of its own.

Because the manuscript is focused mainly on the development of a method, rather than on analysis of a physical system, I feel that a Discussion section would not add clarity to the work, but rather would obfuscate the merits of the method with details about a single application. Instead, **the discussion of benefits and potential drawbacks of the method have been expanded**.

A thought: When we apply specific objective functions, we generally decrease the total variation and find conditions of interest by minimizing the total variance. Would we find similar conditions by not applying any objective functions and maximizing the total variation instead?

The inverse approach is not expected to identify conditions of interest, as shown in Figure 8(b). Maximizing the total variation is not guaranteed to identify any specific condition, but rather identify those that agree with the objective functions the least. For example, instead of applying a linear objective function to the data block to find the cleanest wind speed ramps (as in Figure 10) and selecting the time periods with minimal V is not the same as choosing the maximum V without the application of objective functions.

Are there any available codes or scripts with this method implemented?

No codes or demonstration scripts have been included as part of this submission. Given that the method is relatively straightforward, involving only a handful of welldefined mathematical operations, it seems unnecessary to provide a template for applying the method.

Specific comments:

In the abstract lines 3-4: "Most often, conditions of interest are determined as those that occur most frequently. . ." And similarly, p. 3, lines 12-13: "Within any wind plant data. . ." This statement would benefit from a reference, because it could be argued that the opposite often holds true. E.g. for wind turbine site assessment and certification, the conditions of interest are critical weather and extreme conditions.

The reviewer is correct. Essentially, conditions of interest are necessarily defined by the research, and often times may be focused on infrequent or extreme conditions as these have particular relevance to wind plant behavior and operations. **The phrasing has been changed** to emphasize that in validation of numerical models, commonly occurring conditions are often selected for comparison as they provide the most converged statistics with the least uncertainty due to sample size.

Abstract, "Atmospheric conditions relevant for wind energy research include stationary conditions, given the need for well-converged statistics for model validation, as well as conditions observed less frequently, such as extreme atmospheric events, which are used in wind turbine and wind plant design."

P. 3, "Within any wind plant data, conditions of value for validation are typically identified by way of aggregate statistical metrics or by identifying "wellbehaved" time periods exhibiting a dynamical event or atmospheric condition of interest."

Introduction, p. 2, lines 25-27: This is quite a strong statement – it would benefit from a citation or further argumentation. Introduction, p. 2, lines 27-29: Direct comparison of statistical quantities to what? Why does that discount the coupling between quantities that underpin atmospheric physics? This could be clarified and explained further.

In order to clarify the sentence, the text has been modified and citations have been added to support the statement that,

"Consideration of these variables independently may not provide a complete picture of the state of the atmosphere, as they are inherently correlated (Holtslag and Nieuwstadt, 1986; Kaimal et al., 1976); each variable offers a limited range of insights as to the dynamical state of the atmosphere relevant to the operation of wind energy assets. Further, and perhaps most importantly, consideration of statistical quantities (measures of central tendency, variability, or higher statistical moments) may discount the inherent coupling between quantities of interest that underpin atmospheric physics (Hannesdóttir and Kelly, 2019; Preston et al., 2009; Shahabi and Yan, 2003)."

Figure 1 a): The numbers on the colorbar are missing the number 9 in front of 00 and 50. It is not entirely clear to me what is in error for Figure 1(a). The colorbar appears to be scaled correctly and the labels appropriate. A new figure has been placed in the updated version of the manuscript, but it may be that the pdf rendering through the journal website or pdf viewer may have created some display error.

Equation 4-5: Should just be a single equation with one number. Further, I cannot see how the matrix multiplication would result in the covariance matrix. Unless the average of each column has been subtracted from the values in D I´C and the values have been divided by m. If that is the case, it should be mentioned. At this point in the paper normalization has not been mentioned. Equations 4 and 5 have been combined in the manuscript and a factor of 1/(m-1) has been added for completeness sake. As noted by the reviewer the mean of each channel is removed during the data standardization step, otherwise the correlation matrix would not follow the traditional formulation. The statement about normalization of the data has been moved from Section 4.1 up to the definition of D, where it is more appropriate.

Figure 6: It does not seem that the histogram adds up to 100%. Has the data been cut off at Total Variation=0.3? It would be better to show the whole range of the Total Variation.

The reviewer is correct, and the upper tail of the distribution has been truncated. These distributions include some very high values of V, and have been truncated to emphasize the lower values, where differences between the two distributions are most visible. A note has been added to the caption of Figure 6 clarifying this point, "Both distributions in Fig. 6 have been limited to V≤0.30 to emphasize differences between the two data block lengths. In either case, the distribution is positively skewed, and high values of V exist with very low frequency."

Figure 7 a) is not mentioned anywhere in the text. It should be commented and explained in the text.

Thank you for pointing out this oversight. **Figure 7(a) is now referenced** in the paragraph immediately preceding it where the text is focused around the distribution of observations in the condition space that correspond to the minimum and maximum values of V.

"Fig. 7(a) shows that the periods with minimal values of V have time series that appear constant and experience only small stochastic variations within each channel and that periods with large values of V exhibit more spread"

Page 12, equation 12-13. Again, should just be one equation. Also, what objective function is used for the TI? It is not mentioned.

Equations (12) and (13) have been combined as suggested by the reviewer. The objective function blocks have also been clarified in equations (8)-(10), showing explicitly that in each case, functions are 0 unless otherwise specified.

Page 12: When the objective function eq. 9 is applied to the wind speed, what objective functions are then applied to the direction change and TI at the same time?

In each of the demonstrated regularization schemes, the listed objective function is applied to the specified data channel and the others remain unaltered. That is the other objective functions remain zero. The equations have been modified to highlight the objective function blocks used for regularization, rather than specifying the functions alone. This should make the regularization schemes more clear to the reader.

Page 12, lines 17-18: "Defining specific functions, even of the same forms, would likely increase the average value and spread of V. . .". Are you certain of this? According to Figure 9 a) the average value and spread of V has decreased by subtracting the objective functions from the data. As you mentioned, subtracting the objective function acts as detrending, and therefore it should be expected that the total variation would

always decrease, as it is only the stochastic part of the data that determines the covariance of the remaining data.

The reviewer is correct, general detrending the data should reduce the resultant value of V. The intent of this statement was to convey the idea that if you remove the wrong trend from a time period, you may inadvertently increase V. This is not expected to be the case when using the least-squares minimization to determine fit coefficients as in the article. When the coefficients are prescribed a priori, there is no guarantee that the covariance would be reduced by removing the objective function. The offending sentence has been edited to read,

"Defining the coefficient values ahead of time would likely increase the average value and spread of V; for example, it is not expected that a wind speed ramp with specific slope and vertical offset would fit every time period well, and thus would not necessarily reduce the total variation for that period."

Figure 9: Includes two subfigures named (d). Also, these are not mentioned in the text, but should be. What is fit frequency - is it connected to eq. 11? Could you elaborate? Thank you for pointing this out. The journal prefers subfigures to be collected into single image files, and this was overlooked. The fit frequency refers to the coefficient \$c\_0\$ from equation 10. **The captions in Figure 9 have been updated** to more clearly communicate what information is shown in the distributions.

Section 5: The data used in this section is synthetic, and provides a very illustrative example of the sensitivity. However, I wonder if the removed points can be interpreted as outliers. Could we not say that these are extremes? Maybe the outliers could be assigned standard deviations outside of the range [0, 10], to ensure that they represent "real" outliers due to e.g. measurement errors.

While it is certainly possible for extreme values to excluded as outliers, it should be considered which time periods will be identified as favorable via total variation. If no objective functions are supplied, the method is tuned to quantify the variability of the data about stationary conditions. Extreme values occurring during a given period will probably increase the respective value of V, but these periods should probably not be considered as stationary in any case. If the intent of quantifying V is to identify conditions that include extreme events (gusts, turbulent structures, weather fronts, etc.) the objective functions should be defined to highlight them. Use of the Mahalanobis distance assumes in the current work assumes that each variable is normally distributed within a given time frame. Accordingly, a Mahalanobis distance of 3 implies that there is approximately 1.1% probability of a point being an outlier for two degrees of freedom (as in the outlier sensitivity study) and 2.9% for three degrees of freedom (as in the demonstration with atmospheric variable data). The particular value of the Mahalanobis distance threshold used, should take into account the number of degrees of freedom (i.e. the number of variables) considered in the data. A note to that effect has been added in Section 5. "Any point with  $\chi$  >3 is flagged as an outlier and eliminated. With two degrees of freedom (variables in the data block), values of x >3 are expected to be observed with a probability of approximately 1.1% (Penny, 1996; Ben-Gal, 2005; Gellert et al., 2012)."

Page 16, line 25: ". . . the method is independent of the length of the data record. . .". How can this statement be supported by the current analysis? This statement is intended to communicate that the method does not explicitly require a record of a particular length or resolution. The sentence has been revised to read,

"In addition, the method should be equally applicable to any data, regardless of which variables are part of the data block and for data of any length and resolution, provided that enough observations are present to ensure reasonably converged statistics."

---

## Author Comment (AC2) · 24 Dec 2019

**Response to Comments from Reviewer 2**

amt-2019-200 Total variation of atmospheric data: covariance minimization about objective functions to detect conditions of interest Nicholas Hamilton

There is some good content and work here, with a generalized method to find conditions of interest for multivariate timeseries; and (perhaps more importantly) inclusion of responsible application of a metric (Mahalonbis distance) to evaluate sensitivity of the method to outliers.

Thank you for taking the time to review my submission. I appreciate your concise and direct comments and, in addressing them, I think you will see that the manuscript has been greatly improved. It pleases me that the intended message of the work has been clearly understood and well received. I have provided a brief response to each of the points you raised in the review of my work and, where appropriate, also included any additions or subtractions from the manuscript.

The title is perhaps not quite appropriate; "Total variation of atmospheric data" is rather vague and somewhat grandiose, not accurately capturing the essence of the work and connoting more results/applicability than demonstrated.

I think that your suggestion is correct. The title never felt like it was perfectly suited to the content of the manuscript. Accordingly, the title has been changed to, "Atmospheric condition identification in multivariate data through a metric for total variation", which I believe more concisely conveys the intent of the work and communicates its scope as the development of an analysis and quality control method.

Some significant items of note, as a list:

In the abstract, 'periods' of interest is better expressed as 'conditions', both for the sake of validation and for getting conditional statistics (and towards making fair comparisons of statistics given some conditions).

I think that the suggested change from 'periods' to 'conditions' is appropriate. While the method is designed to quantify the total variability within a continuous time period, it is the identification of atmospheric events or conditions of interest that is the real objective.

Stationarity and conditional statistics underpin this written work; these concepts should be integrated (and referenced, as found in various texts for atmospheric flows), at least starting with the literature review.

The reviewer is correct to point out that the concept of statistical stationarity is one of the main concepts driving the current work. From the fundamental turbulence perspective, the term stationarity is not really expected to apply to data from an

inherently dynamical system (the atmosphere) over periods of this duration. However, the term 'stationary' is also familiar to the atmospheric science community, and has now been mentioned explicitly, as suggested by the reviewer. A statement has been added to Section 4 to underpin the importance of stationarity, "Statistical stationarity (i.e. time-independence of statistical quantities) is a common consideration in turbulence and atmospheric science (Chenge and Brutsaert, 2005; Metzger et al., 2007; Vincent et al., 2010, 2011; Guala et al., 2011). Stationarity is not often assumed for wind energy research and modeling applications, although it is rarely quantified or even considered in validation data."

In your literature review, a key method/scheme for event detection (beyond wavelets) appears to be missing: i.e., reference-signal (or ideal signal) approaches based on Hilbert transform, as in Hristov et al (1998, PRL 81 no.23), used in various literature (e.g. Kelly, Wyngaard & Sullivan 2009).

I would like to thank the reviewer for pointing out this method for detection of atmospheric conditions. A statement has been added to the introduction including the above references.

"Another method for parsing atmospheric conditions found in the literature leverages the Hilbert transform, which convolves time series signals with the Cauchy kernel and results in a phase-shifted set of Fourier components. This method has been used successfully to relate ocean wave conditions to atmospheric conditions through the use of a reference signal (Hristov et al., 1998) and has successfully been extended to turbulence modeling (Sullivan et al., 2000; Kelly et al., 2009) and to relate turbulent motions of various scales within the atmospheric boundary layer (Mathis et al., 2009). Previous use of the reference-signal method (Kelly et al., 2009) required the use of a periodic reference signal, which does not lend itself easily to the detection of nonperiodic atmospheric events, and strongly-correlated ocean wave and turbulent velocity data, which are not available for the majority of wind plant data sets."

When you mention "direct comparison of statistical quantities", it appears that you are trying to refer to statistics based on marginal distributions (or marginal statistics), are you not? In statistical parlance, one contrasts between marginal and conditional statistics.

The reviewer is correct, and that sentence was intended to describe comparison of marginal statistical quantities. The sentence in the introduction has been changed to read,

"Consideration of these variables independently may not provide a complete picture of the state of the atmosphere, as they are inherently correlated (Holtslag and Nieuwstadt, 1986; Kaimal et al., 1976); each variable offers a limited range of insights as to the dynamical state of the atmosphere relevant to the operation of wind energy assets. Direct comparison of the marginal distributions of atmospheric variables aggregates observations without regard to the value of other, potentially correlated variables. Even the use of conditional statistical distributions or measures discounts any dynamic coupling between them and may not fully describe the nature of the atmospheric physics (Hannesdóttir and Kelly, 2019; Preston et al.,

**2009;Shahabi and Yan, 2003)."**

The premise "In lieu of a time series of Richardson number or the Monin-Obukhov stability parameter, turbulence intensity (TI) is used in the current demonstration as a proxy for stability" is fundamentally problematic. That is, the balance of mechanical (shear) production, buoyant production or destruction, and dissipation  $\varepsilon$  (defining the 'simple' conditions where Monin-Obukhov similarity applies) results in TI being a proxy for stability only for flows/conditions with the same dissipation rate (Kelly, Larsen, Dimitrov & Natarajan, 2014). So your results per TI are conditional on  $\varepsilon$ , and do not act as such a proxy unless further constrained (e.g. via U assuming surface-layer similarity for  $\varepsilon$ .) Since stability is not really used in the paper, I suggest that you simply keep TI, and change the justification for its use:  $\sigma$ u and TI are important for driving turbine loads (e.g. Dimitrov, Kelly, Vignaroli & Berg 2018).

Thank you for your concise description of the issue of regarding TI as a proxy for metrics of atmospheric stability. This is an important point to consider when making decisions as to how one should quantify the state of the atmosphere considering the data available. In the current case, as noted by the reviewer, stability is not discussed outside of the referenced section, given that temperature and/or heat flux information are not available for the data used in the current demonstration, it would probably be better to focus the narrative around TI as a relevant quantity of interest for wind turbine loads and wake modeling. The previous framing of the discussion arose from the intent to state that stability is an important factor in describing the state of the atmosphere, while conceding that TI is the quantity considered in many wind energy applications. The relevant excerpt has been changed to read,

"Data used in the current work does not contain any observations of the temperature or heat flux between the atmosphere and the ocean surface, and thus no estimate for the traditional stability metrics are available. Turbulence intensity (TI), although an imperfect proxy of atmospheric stability from a fluid mechanical or atmospheric perspective, provides some sense of the energy contained in the fluctuating flow field, and is well-suited for presenting the utility of the total variation method below. Additionally, TI is a quantity frequently used in the wind energy community to characterize wind plant operating conditions and structural loading of wind turbines (Kelly et al., 2014; Dimitrov et al., 2018) and is often accessible through instrumentation on met masts or wind turbine nacelles making it an appropriate choice for the current demonstration."

In section 3, where you write "without explicitly considering the evolution of atmospheric variables" you should mention stationarity as well. In the atmospheric sciences and boundary-layer meteorology this is typically considered, whereas it is often neglected in wind energy applications.

A similar point from the reviewer regarding the discussion of statistical stationarity has been addressed above. A brief statement has been added to Section 3, noted by the reviewer, reading,

"Considering atmospheric variables in terms of either their marginal distributions (as in Fig. 2 or their conditional distributions (as in Figs. 3 and 5) falls short of saying anything about the dynamics embedded in those observations. Steady-state wake models are defined to represent the timeaveraged flow behind a wind turbine and higher-fidelity models assume that the bulk flow speed and direction do not change in time. Effective validation of numerical modeling tools for wind energy requires that observations conform to stationary atmospheric flow (Chenge and Brutsaert, 2005; Metzger et al., 2007; Vincentet al., 2010, 2011; Guala et al., 2011) or represent a dynamic event of interest."

**Figure 5: missing axis values/scales**

I must apologize for the rendering of the figure. I believe that the axis labels were not included in the typeset document for some reason. In the revised version of the manuscript, Section 3, describing the statistical view of atmospheric conditions, has been reduced in length. Because the 3D histogram did not add significantly to the discussion of the distributions of atmospheric variables beyond the 2D histograms, the figure and associated discussion has been removed.

Section 4: can you interpret the total variation in terms of the multivariate components, to avoid obfuscation? Section 4.0 (p.8) is essentially taken from PCA; you should include reference to appropriate PCA text(s) and try to explain V for the reader. E.g., for readers not as 'fluent' in statistics, if the PC's (P) are orthogonal, then how are the covariances accounted for?

The formulation leading to the total variation does include an eigendecomposition of the covariance matrix and is in fact related derived from PCA. The method was defined this way because PCA was one of the methods originally considered during the analysis. Because the principal components are not identical to the original variances, they must include information from the covariances. That said, the sum of the principal components is also equal to the trace of the covariance matrix, which remains difficult to relate to the covariances between variables. In subsequent work, I found that the determinant of the covariance matrix also reduces the covariance matrix to a single metric that quantifies its variability. In fact, for the current study, the determinant method and the PCA method rank the variability of continuous time periods in the same order, although the numerical value is a bit different. The formulation has been updated using the determinant method, which also happens to be a more direct means at arriving at \$\mathcal{V}\$.

"The total variation, V, of a given regularized data block, D, is expressed as the determinant of the respective correlation matrix,

**$V = det(C) \tag{6}$**

Larger values of Vindicate that the data points are more dispersed in the condition space. In the observational data of the atmosphere discussed here, V>0. The case of V= 0 would indicate that the full n-dimensional condition space is not occupied and some of the variables are perfectly correlated with, i.e. linearly dependent on, some of the others. Metrics of the variation of a multivariable dataset have some history in the literature. Notable past contributions include the pooled10variance method to estimate population variance from those of distinct samples Ruxton (2006), and the 'total' or 'overall' variability Goodman (1968); Anderson (1962) which combine variances of individual variables either linearly or in a sum of squares sense. The generalized variance (Wilks, 1932; Sengupta, 2004), shares a common formulation withV, but has historically been applied to a p-dimensional

**random vector. In contrast, the total variation merges n distinct variables, whose relationship need not be known a priori, and seeks the determinant of the associated correlation matrix"**

Is your V different than the 'overall' or 'total' variability found in literature? It could help also to point out the difference between summative variance and V.

These are good points and, given their similarity, I have decided to answer together. I take it that the reviewer is suggesting that the total variation method be more clearly related or disambiguated from other statistical measures of variability. The metrics total variability, overall variability, and summative variance in common use have slightly definitions and interpretations from the total variation introduced in the current work. Briefly,

**Total variability** is defined as the sum of squares total of difference between expected or mean value and observed qualities.

**Overall variability** refers generally to the variance or standard deviation of a population (i.e. a group of samples considered together).

**Summative** or **pooled variance** refers to the inferred variance of a population of observations from the collection of sample variances.

In contrast, the total variation used in the current work reduces the covariance between normalized variables to a single value through the determinant of the covariance matrix.

A close analog to this method is the generalized variance of a multi-dimensional random vector. Generalized variance was introduced by Wilks as a scalar measure of overall multidimensional scatter. However, in most formulations of generalized variance, the data are considered as a p-dimensional vector. The current work uses the same mathematical operations but applies them to distinct variables that have been merged into a matrix. Mechanically, the same operations are being applied to the data, but given the distinction in formulation, I have elected to maintain the current jargon of 'total variability'. A statement has been added to the introduction with references to some other metrics of variability.

"The metric used to quantify the overall variability of the atmosphere within any given time period is closely related to the generalized variance as per Wilks (1932); Sengupta (2004), but is distinct in that it is applied to a collection of variables rather than a multi-dimensional vector."

Figure 8: suggestion: use logarithmic scale on y-axis to compare more sensibly I thank the reviewer for the suggestion, although I'm not sure I entirely understand what the purpose of logarithmic scaling would be. The figure displays the atmospheric variables considered during time periods with minimum or maximum values of V Given that the data do not span multiple orders of magnitude, rescaling the axes is not expected to add to the interpretation of the data.

Fig.9c: which "dimensionless slope" are you using here?

The dimensionless slope referenced in the caption of Figure 9c refers to the coefficient c\_0 in eq. (7). While all of the coefficients in relationships seen in eqs. (7) - (9) are dimensionless due to the normalization of the variables, the phrasing is a bit difficult to follow. All of the subplots captions have been updated accordingly.

Fig.11: captions are swapped between (c) and (d).

Thanks for catching this oversight. The figure captions have been updated.

Please also note the supplement to this comment:

Additional (minor) comments found in the marked-up document have all been addressed in the manuscript. Thank you for the detailed review of the work. I feel that it is substantially improved due to your thoughtful comments.

---

## Referee Report (RR1)

The manuscript is much improved by the review process and I suggest it should be published after these minor revisions:

- The sensitivity to outliers should be mentioned in the abstract.

- When asking the author if codes with the method implemented were available the reply was: "*No codes or demonstration scripts have been included as part of this submission. Given that the method is relatively straightforward, involving only a handful of well- defined mathematical operations, it seems unnecessary to provide a template for applying the method.*" I would argue that since the method is straightforward, all the more reason to include a demonstration script. A simple script is easier for the readers to follow. I also suggest the author upload the synthetic data that is used in the sensitivity analysis. This also follows the guidelines and recommendations of the journal.

Finally, two points regarding references that I missed in the first review:
- There is a citation to a discussion paper. It is better to cite the final version as written below (and to include the accent over the authors first name ;-)
Hannesdóttir, Á. and Kelly, M.: Detection and characterization of extreme wind speed ramps, Wind Energ. Sci., 4, 385–396, https://doi.org/10.5194/wes-4-385-2019, 2019.

- The IEC 61400-1, p. 177, 2005 citation seems to be wrong (p.4 line 19). There is no mention of cup anemometers in this document and it is only 92 pages. Perhaps another IEC standard should be referenced here?

---

## Author Response (AR2)

**Response to Comments from Reviewer 1 (v2)**

**amt-2019-200**
**Atmospheric condition identification in multivariate data through a metric for total variation**
Nicholas Hamilton

The manuscript is much improved by the review process and I suggest it should be published after these minor revisions:

> *I'd like to thank the reviewer for the thoughtful and thorough reviews for this manuscript. The overall work has been greatly improved thanks to your dedication. Please see responses to each of your comments below.*

The sensitivity to outliers should be mentioned in the abstract.

> *This is an excellent suggestion, thank you. The outlier detection and sensitivity of the method is now mentioned in the abstract.*
> *"**Total variation is somewhat sensitive to the presence of outliers in the input data, and the method is best complemented by quality control procedures to ensure reliable results.**"*

When asking the author if codes with the method implemented were available the reply was: "No codes or demonstration scripts have been included as part of this submission. Given that the method is relatively straightforward, involving only a handful of well-defined mathematical operations, it seems unnecessary to provide a template for applying the method." I would argue that since the method is straightforward, all the more reason to include a demonstration script. A simple script is easier for the readers to follow. I also suggest the author upload the synthetic data that is used in the sensitivity analysis. This also follows the guidelines and recommendations of the journal.

> *Following the reviewer's suggestion. I have made a open-source code repository for the use of interested readers. In the repository are Python codes that can be used to generate the synthetic data as well as real data from a met mast at the National Renewable Energy Laboratory. These are not the data used in the manuscript, but those were supplied by an industry partner and are covered by an NDA. The conclusions section now includes a statement, "**An example Python library has been uploaded to a public repository at https://github.com/nhamilto/total-variation (Hamilton, 2020). In the repository also can be found a synthetic data generator used for the analysis of outlier sensitivity and met mast data similar to that used in the work above. Data in the example is derived from a meteorological mast located at the National Renewable**"*

*Energy Laboratory's Flatirons campus. Additional data from the met mast is available at https://nwtc.nrel.gov/MetData (NREL, 2020) and the atmospheric conditions at NREL are summarized in Hamilton and Debnath(2019)."*

Finally, two points regarding references that I missed in the first review:
There is a citation to a discussion paper. It is better to cite the final version as written below (and to include the accent over the authors first name ;-)
Hannesdóttir, Á. and Kelly, M.: Detection and characterization of extreme wind speed ramps, Wind Energ. Sci., 4, 385–396, https://doi.org/10.5194/wes-4-385-2019, 2019.

> *The reference above has bee updated to refer to the published journal article (and the accents have been included!) Thank you for pointing this out.*

The IEC 61400-1, p. 177, 2005 citation seems to be wrong (p.4 line 19). There is no mention of cup anemometers in this document and it is only 92 pages. Perhaps another IEC standard should be referenced here?

> *The reviewer is absolutely correct in this point. This reference has been updated to the IEC 61400-12-1 Standard for Power Performance Measurements of Electricity Producing Wind Turbines. The citation in text has been updated and the standard now appears in the references section.*

**Response to Comments from Reviewer 2 (v2)**

**amt-2019-200**
**Atmospheric condition identification in multivariate data through a metric for total variation**
Nicholas Hamilton

The article has improved somewhat upon revision, though there are still some things that need to be addressed; I list them here.

*I'd like to thank the reviewer for the thoughtful and thorough reviews for this manuscript. The overall work has been greatly improved thanks to your dedication. Please see responses to each of your comments below.*

First, the reason that the Hilbert-transform (Hristov) method was suggested, was because it does _not_ require a periodic reference signal. The author has now included references to this method and application, but however incorrectly followed with "Previous use of the reference-signal method (Kelly et al., 2009) required the use of a periodic reference signal, which does not lend itself easily to the detection of non-periodic atmospheric events".

*Thank you for pointing out this oversight. I must admit that upon review of the work cited in the comment above, the I was unable to find any discussion of a periodic reference signal. This mixup must have come from another article that I was considering including in the literature review. The statement above has been removed from the manuscript to remove any misleading statements.*

The author gives a nice response to the reviewer's question about the difference between _V_ and similar measures found in the literature. He responds including distinguishing between total variability, overall variability, and summative or pooled variance; he also writes about generalized variance and its relation. However, the revision (end of §1, pg.3) doesn't include such, only mentioning that V "is closely related to the generalized variance as per Wilks (1932); Sengupta (2004), but is distinct in that it is applied to a collection of variables rather than a multi-dimensional vector." To help the reader understand, this response should be better integrated into the article (some is on p.8).

*The more complete comparison of the current method to previous work in the literature was excluded only for brevity. As recommended by the Reviewer, the explanation included in the previous response document has been added to the manuscript.*

*"**Alternate metric sexist that quantify the variability of multiple samples or multivariate data. The metrics total variability, overall variability, and***

*summative variance in common use have slightly definitions and interpretations from the total variation introduced in the current work. Briefly, total variability is defined as the sum of squares total of difference between expected or mean value and observed qualities.Overall variability refers generally to the variance or standard deviation of a population (i.e.a group of samples considered together).Summative or pooled variance refers to the inferred variance of a population of observations from the collection of sample variances. In contrast, the total variation used in the current work reduces the covariance between normalized variables to a single value through the determinant of the covariance matrix. A close analog to this method is the generalized variance of a multi-dimensional random vector. Generalized variance was introduced by Wilks (1932) and Sengupta (2004) as a scalar measure of overall multidimensional scatter. However, in most formulations of generalized variance, the data are considered as a p−dimensional vector. The current work uses the same mathematical operations but applies them to distinct variables that have been merged into a matrix. Mechanically, the same operations are being applied to the data, but given the distinction in formulation, the current work adopts the jargon of 'total variability'.*"

The addition in section 3 (lines 5-10, p.5) is somewhat problematic.
Line 5: need close-parens after 'Fig.2'.
    *Corrected. Thank you.*

Pg.5, line6: the added text "Steady-state wake models..." should be preceded by 'E.g.,'. Why introduce this sentence? It's a bit awkward, and also some 'high-fidelity models' do not assume unchanged speed or direction. Following that sentence, the author states that effective validation requires stationarity or that "observations represent a dynamic event of interest", but simple measures to deal with nonstationarity (e.g. detrending) are also sometimes used.
    *The sentence has been amended to read,*
    *"**For example, many steady-state and analytical wake models are defined to represent the time-averaged flow behind a wind turbine and many uses of high-fidelity models assume that the bulk flow speed and direction do not change in time.**"*
    *This statement summarizes one of the key motivations from the work, that is to identify more or less stationary atmospheric conditions in data sets used for model validation. Most steady-state and analytical wake models (FLORIS, FAST.Farm, WaSP, etc.) assume stationarity. Validation should select data that reflect the design scope of the models, otherwise there is little reason to expect good agreement with observational data.*

p.7, line 24: is it literally normalization by "respective span" (max minus min), or by respective variance?

*Thank you for pointing this out. That statement should have been updated during the previous revision. The sentence now reads,*
***"Each variable has been normalized by its standard deviation…"***

p.8, line 20: '(' should preceed 'Ruxton'.

*Corrected. Thank you.*

p.16, l.18-20: the DLC's of the IEC 61400-1 do not fit actual events, but are mostly calibrated statistically for the sake of loads, based on expected joint occurence of various events and scenarios (e.g. Hannesdottir & Kelly, Dimitrov et al).

*To clarify, The intent of this statement in the conclusions is to suggest that the definition of extreme atmospheric events (say extreme operating gusts (as in section 6.3.3.3 in the IEC standard) could be used in the function block to regularize the data. This would provide an algorithmic means of detecting extreme events in historical data records and offer researchers a means of grouping specific periods for further analysis. The statement has been updated accordingly,*

[revised manuscript text omitted]